# Is Obesity a Cause for Shame? Weight Bias and Stigma among Physicians, Dietitians, and Other Healthcare Professionals in Poland—A Cross-Sectional Study

**DOI:** 10.3390/nu16070999

**Published:** 2024-03-29

**Authors:** Alicja Baska, Karolina Świder, Wojciech Stefan Zgliczyński, Karolina Kłoda, Agnieszka Mastalerz-Migas, Mateusz Babicki

**Affiliations:** 1Department of Lifestyle Medicine, School of Public Health, Centre of Postgraduate Medical Education, 01-813 Warsaw, Poland; wzgliczynski2@cmkp.edu.pl; 2Polish Society of Lifestyle Medicine, 00-382 Warsaw, Poland; 3NZOZ Biogenes, 53-224 Wroclaw, Poland; karolina.swider.pl@gmail.com; 4MEDFIT Karolina Kłoda, 70-240 Szczecin, Poland; wikarla@gazeta.pl; 5Scientific Section of the Polish Society of Family Medicine, 51-141 Wroclaw, Poland; agnieszka.migas@gmail.com (A.M.-M.); ma.babicki@gmail.com (M.B.); 6Department of Family Medicine, Wroclaw Medical University, 50-367 Wroclaw, Poland

**Keywords:** weight bias, stigma, obesity, overweight, healthcare professionals

## Abstract

Weight bias and weight stigma pose significant challenges in healthcare, particularly affecting obesity management practices and patient care quality. Our study evaluates their prevalence and impact among healthcare professionals in Poland. Using the Fat Phobia Scale and custom questions, we surveyed 686 professionals via Computer-Assisted Web Interview (CAWI). Results reveal a moderate level of explicit weight bias (mean score: 3.60 ± 0.57), with significant variations across professional groups: physicians (3.70 ± 0.48), dietitians (3.51 ± 0.48), and others (3.44 ± 0.77). Common feelings towards individuals with obesity include willingness to help (57.0%) and compassion (37.8%), yet 29.9% perceive obesity as shameful. The results also vary depending on the respondent’s sex or BMI. These findings underscore the need for evidence-based interventions to mitigate weight stigma and enhance understanding of obesity among healthcare professionals.

## 1. Introduction

Weight bias is described as the presence of negative weight-related attitudes, belief assumptions, and judgments held about people living in large bodies [1]. It encompasses three constructs: prejudice (as a negative evaluation of a social group/individual), stereotyping (convictions about the etiology and/or maintenance of obesity), and discrimination (actions or behaviors) [2]. The frequent examples of weight bias include attributing the following to individuals with obesity: laziness, lack of willpower, low level of intelligence, a lack of moral character, bad hygiene, and unattractiveness [3]. The World Obesity Federation distinguishes weight (or obesity) stigma from weight bias as its manifestation, defined as discriminatory acts and ideologies targeted towards individuals because of their weight and size [3]. Weight stigma can manifest as negative comments directed at people with obesity, reinforcement of harmful stereotypes in the media, and environmental factors, such as medical settings that may not adequately accommodate individuals with obesity.

There is a mounting body of evidence that demonstrates how weight bias and stigma pose a significant challenge to the effective treatment and prevention of obesity. This challenge stems from their detrimental impact on health and their widespread prevalence, including in healthcare settings [4]. In the “Joint international consensus statement for ending stigma of obesity” published in Nature Medicine in 2020 [5], it was concluded that weight stigma has significant adverse effects on both mental and physical health. Weight-based stigma is associated with increased risks of depressive symptoms, anxiety, substance use, and social isolation, as well as reduced self-esteem and heightened stress levels [5]. A 2018 systematic review found that obesity stigma is positively linked to various health indicators, including conditions such as obesity itself and diabetes, as well as physiological markers like cortisol and C-reactive protein level and oxidative stress [6]. This underscores the far-reaching impact of weight stigma on health outcomes. Additionally, there is evidence suggesting that individuals who experience weight-based stigma are less likely to adopt healthy behaviors, such as engaging in physical activity and maintaining a healthy diet. This, in turn, heightens the risk of exacerbating obesity [7]. Moreover, patients with obesity may encounter barriers in accessing appropriate healthcare, not only in relation to obesity but also for other health conditions and preventive care, such as cancer screenings. These barriers include experience of previous stigma from healthcare professionals, feelings of embarrassment, guilt, or fear related to the equipment not being able to accommodate individuals with obesity [8]. It is also important to emphasize that the assumption that obesity is a matter of choice and can be reversed through voluntary decisions, such as ‘eating less and moving more’, not only harms the individuals affected but also has implications for public health and research [5]. The impact of this assumption is visible, for example, in public campaigns and policies that tend to overlook environmental and societal factors that play a major role in the epidemic of obesity, which in turn can further perpetuate obesity stigma among individuals [9].

The most recent Canadian Adult Obesity Clinical Practice Guidelines published in 2020 included weight bias, stigma, and discrimination for the first time in its history [1]. These issues were addressed in a separate initial chapter “in recognition of emerging and compelling evidence that they represent a significant challenge to practice and policy”. The primary recommendation in the Canadian guidelines calls for healthcare providers to assess their attitudes and beliefs regarding obesity and consider how these attitudes and beliefs may influence care delivery (Level 1a; Grade A). Various measures are available to assess weight bias, validated and widely used in research, including the Fat Phobia Scale (FPS), the Antifat Attitudes Test (AFAT), the Antifat Attitudes Scale (AFAS), and others [10], sometimes in combination.

Weight bias has been reported among healthcare professionals of different professions, including primary care physicians, nurses, dietitians, and medical students worldwide. One study examining weight stigma among women who were overweight and obese found that as much as 53% of them experienced inappropriate comments from their physicians, who ranked as the second most common source of weight stigma (out of 20 different sources). Participants reported similar experiences with nurses (46%) and dietitians (37%) [11]. It is known that weight bias in healthcare settings reduces the quality of care, impacting not only obesity management practices but also the patient–healthcare professional relationship, consequently affecting patients’ engagement in healthcare services [12,13].

In 2022, the Polish Society for the Treatment of Obesity published guidelines that also included a chapter on countering discrimination and stigmatization of patients with obesity [14], followed by the release of the “Charter of Rights for Patients with Obesity” [15]. Despite these developments, scientific research on the prevalence and characteristics of weight bias among healthcare professionals in Poland has been limited, and most of the few studies available only used non-validated, custom questionnaires, making the results difficult to compare and evaluate. As a response to this gap, our research group, comprised of representatives from the Scientific Section of the Polish Society of Family Medicine in collaboration with the Centre of Postgraduate Medical Education and the Polish Society and Lifestyle Medicine, embarked on a study focusing on weight stigma and fat phobia in Poland using the Fat Phobia Scale (FPS) combined with a custom questionnaire. This article represents the second part of our research, with the first part primarily investigating weight bias among the general public [16]. Therefore, the aim of this study was to evaluate the level of weight bias and stigma among healthcare professionals (physicians, dietitians, and others) in Poland.

## 2. Materials and Methods

The study was conducted using a Computer-Assisted Web Interview (CAWI). The data was collected from April to July 2023. Prior to completing the questionnaire, participants were presented with information about the research aim and methodology and were required to provide informed consent. The data collection process ensured full anonymity; no personal data was collected.

The survey was distributed among Polish resident doctors attending obligatory specialization courses organized by the Centre of Postgraduate Medical Education as well as via social media platforms, including Facebook and Instagram, specifically targeting medical professionals, such as groups for medical doctors who require confirmation of a medical license. An invitation to complete the questionnaire was also shared via newsletters of the Polish Society of Family Medicine and the Polish Society of Lifestyle Medicine.

The research tool was an online questionnaire prepared in Polish via Google Forms. It consisted of four sections. The first section collected information on the characteristics of the respondents (gender, age, place of residence, weight, height, and profession). The respondents were also asked to indicate the frequency of contact with people living with obesity. The second part used the Fat Phobia Scale (FPS), examining explicit weight bias. The scale consists of 14 pairs of adjectives (positive/negative) designed to characterize a person with obesity, using a five-point Likert scale. The total scale score was obtained by summing the scores from each question and dividing by 14 (the number of scale items), resulting in a value ranging from 1 to 5. A higher score on the scale indicates a greater level of fat phobia. According to the scale design, a score of 2.5 indicates a neutral attitude. Scores above this value suggest a negative attitude, while scores below indicate a positive attitude. A score of 4.40 or above indicates a high level of fat phobia, while a score of 3.6 is seen as an indication of an average amount of fat phobia. The internal consistency of the tool was found to be 0.893 [17]. The FPS was translated into Polish by a certified bilingual team member. It was then independently back-translated into English by two medical professionals proficient in English. The original FPS was verified against the back translations, and no significant differences were found. The third section included custom questions assessing the level of stigma, describing the feelings of the respondents when in contact with people with obesity (for example, the likelihood of dating, hiring, entrusting a child in their care, and befriending a person with obesity). The fourth section comprised custom questions concerning knowledge about obesity. These questions aimed to verify whether respondents perceived obesity as a chronic disease and whether they could identify the BMI category for diagnosing obesity (single-choice questions). The questions also assessed their knowledge of obesity causes, effective forms of treatment, and obesity-related conditions (multiple-choice questions). The custom questions in both sections were developed by our multidisciplinary team of authors, drawing on their many years of expertise in obesity treatment, public health, medical education, medical communications, patient advocacy, and extensive literature review.

### Statistical Analysis

The analysis was carried out using Statistica 13.0 by StatSoft (Cracow, Poland). The analyzed variables were of qualitative and quantitative nature. The normal distribution was assessed using the Shapiro–Wilk test. Basic descriptive statistics were applied, including percentages, means, and standard deviations. The comparison of qualitative variables was conducted using the chi-square test. For quantitative variables, non-parametric tests such as the Mann–Whitney U test or the Kruskal–Wallis Test were used. The degree of correlation between quantitative variables was evaluated using the Spearman correlation test. The significance level was assumed at 0.05.

## 3. Results

### 3.1. Characteristics of the Study Group

The study involved 686 healthcare professionals, including 404 physicians (58.7%), 117 dietitians (17.0%), and 167 individuals representing other medical professions (24.3%). The latter group included 42 nurses (6.1%), 22 physiotherapists (3.2%), 12 pharmacists (1.7%), 12 dentists (1.7%), 3 paramedics (0.6%), and 76 people other than those listed (11.0%). The average age of the respondents was 34.6 ± 7.7 years. The majority were females (88.1%) and residents of large cities (69.9%). The majority of respondents exhibited a normal BMI (60.5%). Notably, there were significant differences in the prevalence of overweight and obesity among different members of the studied group, with physicians recording rates of 18.6% and 17.3%, dietitians 10.3% and 4.2%, and other health professionals 26.4% and 18.6%, respectively. A detailed overview of the characteristics of the study group is presented in Table 1.

### 3.2. Fat Phobia Scale (FPS)

In the examination of the FPS scale, respondents achieved an average score of 3.60 ± 0.57, suggesting an average level of fat phobia [17]. The highest scores were recorded for the assessment of “dislikes food/likes food” (4.13 ± 0.89) and “undereats/overeats” (4.09 ± 0.87). Conversely, the lowest scores were noted for the comparisons “weak/strong” (3.00 ± 0.86) and “lazy/industrious” (3.16 ± 1.68).

When comparing the level of fat phobia among various groups of healthcare professionals, physicians scored higher on all the individual items and exhibited the highest level of explicit weight bias (3.70 ± 0.48), surpassing that of both dietitians (3.51 ± 0.48) and a group comprising other healthcare professionals (3.44 ± 0.77). Both the group of dietitians and the group of other healthcare professionals demonstrated a below-average level of fat phobia (*p* < 0.001). A detailed analysis of the FPS scale scores can be found in Table 2.

It was also demonstrated that a negative correlation exists between the FPS score and the likelihood of engaging in various social interactions with people with obesity. The strongest association was observed for the probability of going on a date with an individual living with obesity. A detailed analysis of the correlation between the FPS score and social interactions with individuals with obesity is presented in Table 3.

### 3.3. Custom Questions Assessing the Level of Stigmatisation

In analyzing custom questions evaluating the perception of patients with obesity, it was found that 94.2% of medical professionals believe that people with obesity are not inherently worse than those with normal weight. Moreover, 70.1% indicated that obesity is not a reason for shame. However, a notable 74.6% of respondents expressed the view that individuals with obesity are less attractive than those with normal body weight. This perception varied, with 80.0% of physicians holding this belief compared to 66.6% of dietitians and 67.0% of other medical professionals (*p* < 0.001). Furthermore, when assessed on a 10-point scale, the likelihood of going on a date with a person with obesity received the lowest score of 4.47 ± 2.89 compared to other social interactions, while befriending a person with obesity emerged as the most probable (8.91 ± 1.94).

Among the respondents, the most prevalent feelings accompanying interactions with individuals living with obesity included the willingness to help (on average, 57.0% of respondents), kindness (47.4%), and compassion (37.8%). The least frequent were contempt (2.6%), mercy (8.6%), and reluctance (8.7%). Notably, differences between the two homogenous groups were observed. In comparison to dietitians, physicians seemed to experience less kindness (44.3% vs. 64.9%, *p* < 0.001), liking/affection (20.1% vs. 31.6%, *p* = 0.023), and willingness to help (63.6% vs. 70.9%, *p* < 0.001), while expressing more impatience (14.9% vs. 2.6%, *p* < 0.001), reluctance (10.6% vs. 2.6%; *p* = 0.009), and discomfort in contact (13.9 vs. 7.7). However, physicians also demonstrated more compassion (42.1% vs. 35.0%, *p* = 0.012). When asked to provide their opinion on whether individuals living with obesity face discrimination using a 10-point scale, respondents provided an average score of 7.62 ± 2.27. There were no major differences between different healthcare professionals’ groups. A detailed overview of the custom questions related to obesity stigma is presented in Table 4.

### 3.4. Other Factors Influencing the Perception of Weight Stigma

#### 3.4.1. Sex

The results indicate that female medical professionals notice higher levels of discrimination against individuals living with obesity in Poland (7.81 vs. 6.23, *p* < 0.001). They are also more inclined to engage in positive social interactions with these individuals (dating, befriending, employing, or childcare provision). Female healthcare professionals report feeling more kindness (49.3% vs. 33.3%, *p* = 0.016) and affection (24.6% vs. 14.8%, *p* = 0.029) while experiencing less contempt (2.0% vs. 7.4%, *p* = 0.048) and indifference (17.3% vs. 34.6%, *p* < 0.001) in interactions with individuals living with obesity. For the remaining 6 out of 10 feelings (mercy, reluctance, discomfort, impatience, compassion, and willingness to help), similar trends were observed, although the differences between male and female respondents were not found to be statistically significant. Similarly, no significant difference in FPS scores between the two sexes was found.

#### 3.4.2. Respondents’ Body Mass Index

Individuals meeting the obesity criteria reported the highest scores when asked about the incidence of discrimination against people living with obesity. Similarly, they also obtained the highest scores in assessing the likelihood of positive social interactions with people living with obesity (dating, befriending, employing, or childcare provision). Healthcare professionals with a BMI indicating obesity report feeling more kindness and compassion than healthcare professionals with a normal BMI while experiencing less reluctance. For the remaining 7 out of 10 feelings (mercy, contempt, discomfort, affection, impatience, indifference, and willingness to help), similar trends were observed, although the differences between respondents with underweight, normal BMI, overweight, and obesity were not found to be statistically significant. Similarly, only slight differences were observed in FPS scores: the respondents with obesity had the lowest FPS score (3.50) compared to respondents with overweight (3.61), normal weight (3.61), and underweight (3.61). These differences, though, were not statistically significant.

#### 3.4.3. Place of Residence

No statistically significant differences were identified based on the place of residence.

Detailed results analyzing the impact of demographic variables and respondents’ BMI are presented in Table 5 and Table 6.

### 3.5. Level of Knowledge about Obesity

In the analysis of questions aimed at assessing the level of knowledge about obesity, it was found that 98.5% of medical professionals acknowledge obesity as a chronic disease requiring treatment, while only 84.7% correctly identify the BMI category for diagnosing obesity. Regarding the reasons for obesity development, respondents most commonly selected excessive calorie intake (98.8%), lack of physical activity (94.3%), or complications from certain medications (91.0%). Notably, 26.5% of respondents incorrectly identified hyperthyroidism as a cause of obesity, with the majority from the group of other medical professionals (58.1%).

When it comes to effective treatment methods, the most frequently chosen answer was regular physical activity (97.5%), followed by pharmacological treatment (87.1%) and bariatric surgery (85.6%). Substantial differences were observed among various respondent groups. Physicians significantly favored pharmacological treatment (93.6%) and bariatric surgery (93.1%) as effective treatments, in contrast to dietitians (75.2%, 75.2%) and other health professionals (79.6%, 74.9%). Surprisingly, 4.1% of respondents considered fasting as an effective form of obesity treatment, with higher percentages among other medical professionals (6.6%) and physicians (4.0%), but only one dietician (0.9%).

Among the most commonly indicated obesity-related conditions, respondents frequently chose hypertension (98.7%), decreased exercise tolerance (98.7%), diabetes (98.4%), depression (98.4%), and hormonal disorders (95.9%). Less frequently mentioned complications included deterioration of hair and/or nail growth (74.7%) and gout (71.7%). Only 63.5% of respondents (65.6% physicians, 77.8% dietitians, and 48.5% representatives of other medical professions) acknowledged that obesity can contribute to the development of dementia.

A detailed overview of the level of knowledge distinguishing between different groups of healthcare professionals is presented in Table 7.

## 4. Discussion

While weight stigma manifests across multiple settings, healthcare settings require special consideration due to their critical role in caring for and treating individuals’ physical and mental health.

Healthcare professionals, participants in our study, attained an average Fat Phobia Scale (FPS) score of 3.60 ± 0.57, indicating an average level of fat phobia. Our analysis revealed discernible variations among different medical professions, with physicians exhibiting the highest weight bias (FPS = 3.70), followed by dietitians (FPS = 3.51) and other healthcare professionals (FPS = 3.44). These findings suggest a moderate level of fat phobia among Polish doctors and a comparatively lower level among other healthcare professionals. To our knowledge, weight bias has not been extensively explored in Poland, and no study has previously examined its extent among healthcare professionals using validated tools. However, a 2020 study in Poland found that 82.6% of patients with obesity experienced inappropriate behaviors, often attributed to healthcare professionals [18]. Additionally, a recent Polish study reported that 48.4% of respondents witnessed discriminatory behaviors of medical staff toward patients with obesity [19].

Weight bias among healthcare professionals in Poland appears to surpass that of professionals in other countries. A large multinational study published in 2020, encompassing over 1,500 healthcare professionals from 77 countries, estimated the FPS score among healthcare professionals to be 3.40 [20], while a 2021 systematic review and meta-analysis—based on five studies utilizing the FPS—estimated a mean FPS score of 3.48. Notably, dietitians demonstrated the lowest explicit weight bias with a score of 3.37, slightly lower than their counterparts in Poland (3.51). In contrast, physicians in Poland exhibited a similar level of explicit weight bias compared to general practitioners in Germany (3.70) [19].

Although weight bias has been reported in multiple studies among representatives of various medical professions, including physicians (e.g., general practitioners [21] or professionals engaged in the management of obesity [22], dietitians [23], nurses [24], and physical therapists [25]), to the best of our knowledge, no other study has specifically compared weight bias among different healthcare professionals using the FPS. However, such a comparison was made among UK trainee dietitians, doctors, nurses, and nutritionists [26]. Interestingly, in contrast to our study, no significant differences in FPS scores between physicians- and dietitians-to-be were found. Nonetheless, their scores were higher than the mean FPS score for all healthcare professionals summarized in the aforementioned meta-analysis and higher than those of the population of healthcare professionals in Poland.

In conjunction with the Fat Phobia Scale, numerous studies worldwide have employed various tools, including the Antifat Attitudes Scale [27] and the Attitudes Towards Obese Persons Scale [25,28], to investigate explicit weight bias among healthcare professionals. Additionally, multiple studies utilizing custom questionnaires and various methods aim to provide valuable insights for a better understanding of the potential consequences of this phenomenon [22]. For example, a study involving medical students presented sample presentations of clinical cases, revealing that patients with obesity were often described as less attractive and compliant compared to patients with normal weight [29]. Similarly, in a study comparing the reactions of physicians, doctors reported feeling more negatively toward patients who were overweight and suggested they would spend less time with them [30]. Among nurses, between 5.9% and 24.3% reported feeling repulsed by individuals living with obesity [31], while 34.6% to 47.7% reported experiencing discomfort when caring for patients with obesity [32]. In our analysis, we attempted to identify the feelings accompanying healthcare professionals in interactions with individuals living with obesity as another way of quantifying weight bias. It revealed that, on average, 29.9% of health professionals in Poland identify obesity as a cause for shame, 19.4% of them experience indifference in contact with people with obesity, 12.4% feel discomfort, and 10.3% express impatience. On the other hand, on average, 57.0% feel a willingness to help, 47.4% show kindness, and 37.8% show compassion.

These biases are reflected in patient experiences, with patients reporting disrespectful, patronizing, insensitive, and/or contemptuous treatment from healthcare professionals [33]. A study conducted in Poland revealed that patients with obesity often experience different forms of improper behavior, including unpleasant, judgmental comments (81%), disdainful remarks (77%), and disgruntled grimaces (68%). They also admitted feeling blamed for carrying excess weight (73%).

Findings from all these studies, spanning as far back as 1989, consistently affirm the enduring existence of weight bias within the healthcare professional community [10] as a problem requiring urgent action [5]. Poland is no exception, especially considering that the levels of weight stigma appear to be mildly higher than in other studied populations.

The assessment of weight bias often involves an exploration of contributing factors. As multiple studies conducted among healthcare professionals highlighted the relevance of the respondents’ own BMI, in our analysis, we also looked into the correlation between respondents’ self-reported BMI and attitudes towards individuals living with obesity. We found that healthcare professionals with a BMI indicating obesity scored the highest in the question regarding the incidence of discrimination against people with obesity in Poland. They also often experienced more kindness and compassion and less reluctance in interactions with individuals with obesity compared to respondents with normal BMI. These differences were not reflected in the level of weight bias assessed by the Fat Phobia Scale. This contrasts with several studies where higher self-reported BMI was predictive of lower fat phobia, including studies conducted among physicians [34], health professionals specializing in obesity [22], and trainees [26].

Our study also revealed significant differences in weight bias and attitudes across genders. The female respondents declared experiencing more kindness and affection while less contempt and indifference in interactions with people living with obesity compared to men. They were also more likely to engage in positive social interactions and seemed to be more sensitive to noticing discrimination. Several studies have investigated gender differences in weight bias, consistently highlighting lower bias in women. A 2012 study [34] involving a substantial sample of medical doctors in the US (nearly 2300) revealed that females exhibited less implicit and explicit bias. Similarly, two studies conducted among general practitioners in Germany [21] and Canada [35] found that female physicians held fewer negative attitudes than their male counterparts. Another study involving healthcare professionals from over 70 countries [20] reported similar trends. However, a study focusing on healthcare professionals specializing in obesity [22] reported that women expressed significantly stronger implicit bias than men, while measures of explicit weight bias did not show an association with gender. Likewise, in our study, explicit weight bias measured by FPS did not exhibit significant differences between genders.

Numerous studies also propose that age serves as yet another significant factor influencing the level of weight bias. This connection was identified in the previously mentioned study conducted among healthcare professionals specializing in obesity [22]. Additionally, a similar correlation is hinted at by the findings in studies involving German general practitioners and those conducted among medical and nutrition students [21,26]. However, in our study, the analysis of age’s impact on weight bias among healthcare professionals in Poland was constrained by the homogeneity of our respondent group. Consequently, the determination of whether age indeed influences weight bias in this specific population remains uncertain. Further research with a more diverse and experienced participant pool is warranted to provide more comprehensive insights.

In addition to evaluating healthcare professionals’ attitudes, our study also assessed respondents’ knowledge of the disease. An overwhelming 98.5% of medical professionals acknowledge obesity as a disease. Comparatively, in a similar study conducted recently in Sweden among primary care physicians, 91% of respondents recognized obesity as a disease [36].

One question revealed a significant difference between the various groups of our respondents who pertained to effective treatment methods. Physicians overwhelmingly favored pharmacological treatment and bariatric surgery, with 93.6% and 93.1% of them, respectively, indicating these as effective treatment methods in a multiple-answer question. This contrasts notably with the findings from the Swedish study, where over half of the respondents (58%) did not believe that pharmacological treatment was effective. An explanation of these differences might be attributed to the time of data collection: the Swedish study was conducted in 2021, a period when pharmacological treatment might not have been as widely accessible and commonly used, and not all currently available and registered drugs were readily available in Sweden at that time.

Similarly, a 2020 study involving over 1500 healthcare professionals from nearly 80 countries shed light on perceptions regarding effective treatment methods. In a single-answer question asking respondents to indicate the most effective treatment for severe obesity, 61% chose bariatric surgery, while 37% believed that lifestyle interventions (diet and exercise—16%, psychological support and behavior modification—21%) were more effective [20]. To the best of our knowledge, this study is one of the two studies conducted in Poland that compare knowledge on obesity among different healthcare professionals. The earlier study, published in 2023 [37], had a smaller group of 184 respondents and substantially differed in the composition of the study group, including 52.2% of physicians, 20.7% of nurses, and 19.0% of physiotherapists compared to 58.7% of physicians, 17.0% of dietitians, and 24.3% of other healthcare professionals (including nurses, physiotherapists, pharmacists, dentists, and paramedics). Interestingly, no statistically significant differences in knowledge were observed among representatives of different medical professions in that study, in contrast to our research, where several such correlations were found. While both studies observed similar knowledge levels regarding BMI criteria for recognizing obesity (84.7% in our study vs. 89.1% in the previous study), the results concerning methods of treatment cannot be directly compared. This discrepancy arises from the fact that data for the previous study were collected from January 2019 to September 2020, when currently available methods of pharmacological treatment were unavailable. Thus, our study appears to be the only recent investigation assessing perceptions of effective obesity treatment methods among healthcare professionals in Poland.

The authors acknowledge the inherent limitations associated with the methodology employed in this study, particularly regarding the chosen data collection methods. However, leveraging online platforms and social media substantially aids in reaching a diverse audience across Poland. The utilization of an anonymous Internet survey format might encourage more candid responses, especially given the sensitivity of the subject matter, although it does not exclude the social desirability bias. The analyzed sample does not represent the entire population of healthcare professionals in Poland, emphasizing the need for future studies to provide a more comprehensive understanding of the topic. What is more, this cross-sectional study does not provide information on the evolution of the results over time.

The results indicate that constant monitoring of the level of fat phobia and weight stigma among healthcare professionals is necessary. Conducting such surveys may have the added value of increasing awareness of the challenge of attitudes towards patients with excess body weight and thus may positively influence the effectiveness of the treatment process.

## 5. Conclusions

The results of this study reveal a moderate level of weight stigma among healthcare professionals in Poland as well as highlight gaps in knowledge about obesity. Given the well-documented negative impact of weight stigma on healthcare quality, it is imperative to implement evidence-based interventions aimed at enhancing the knowledge, skills, and competencies of health professionals. These interventions should include educating professionals about uncontrollable and non-modifiable causes of obesity, improving communication skills, integrating patient perspectives into educational programs, and fostering self-assessment of attitudes and internalized weight bias.

## 6. Recommendations

We strongly advocate for utilizing these findings to inform institutions, scientific societies responsible for healthcare professional training, and public health entities in Poland. By expanding and leveraging these results through further research in this field, effective, evidence-based interventions can be developed to combat weight bias and stigma and enhance understanding of obesity among healthcare professionals. This collaborative effort will ultimately contribute to improving healthcare delivery and promoting equitable treatment for individuals affected by obesity.

## Figures and Tables

**Table 1 nutrients-16-00999-t001:** Characteristics of the study participants.

Analyzed Variables		Studied Group N(%)/M ± SD	Healthcare Professional			
			Physicians (%)/M ± SD	Dieticians (%)/M ± SD	Others (%)/M ± SD	*p*
Sex	Female	606 (88.1)	348 (86.1)	111 (94.9)	147 (88.0)	0.134 *
	Male	82 (11.9)	56 (13.9)	6 (5.1)	20 (12.0)	
	Other	0 (0.0)	0 (0.0)	0 (0.0)	0 (0.0)	
Age (years)		34.6 ± 7.7	34.7 ± 6.9	31.6 ± 6.7	36.4 ± 9.3	**0.005 #**
Weight (kg)		69.7 ± 16.9	70.5 ± 16.6	62.2 ± 10.9	73.2 ± 19.4	**<0.001 #**
Height (cm)		168.6 ± 7.4	168.7 ± 7.5	167 ± 6.7	168.7 ± 7.6	0.503 #
BMI (kg/m^2^)		24.5 ± 5.4	24.7 ± 5.3	22.1 ± 3.3	25.6 ± 6.3	**<0.001 #**
BMI	Underweight	35 (5.1)	16 (4.0)	9 (7.7)	10 (6.0)	**<0.001 ***
	Normal weight	416 (60.5)	243 (60.1)	91 (77.8)	82 (49.0)	
	Overweight	131 (19.0)	75 (18.6)	12 (10.3)	44 (26.4)	
	Obesity	106 (15.4)	70 (17.3)	5 (4.2)	31 (18.6)	
Place of residence	Rural area	69 (10.0)	32 (7.9)	15 (12.8)	22 (13.2)	**0.001 ***
	Town of up to 20,000 inhabitants	39 (5.7)	16 (4.0)	4 (3.4)	19 (11.4)	
	City of 20,000–100,000 inhabitants	99 (14.4)	51 (12.6)	23 (19.7)	25 (15.0)	
	City of 100,000–500,000 inhabitants	144 (20.9)	89 (22.0)	22 (18.8)	33 (19.8)	
	City of over 500,000 inhabitants	337 (49.0)	216 (53.5)	53 (45.3)	68 (40.6)	
Healthcare professional	Physicians	404 (58.7)	-	-	-	-
	Dieticians	117 (17.0)	-	-	-	
	Others	167 (24.3)	-	-	-	

M—mean, SD—Standard deviation, N—number, kg—kilograms, cm—centimeters, BMI—body mass index, # Kruskal–Wallis H Test, * Chi-squared test, Significant effects (<0.05) are marked in bold.

**Table 2 nutrients-16-00999-t002:** Analysis of the FPS (Fat Phobia Scale) scores considering individual questions and distinguishing between doctors, dieticians, and other respondents.

FPS Pair of Adjectives	Studied Group M ± SD	Healthcare Professionals			
		Physicians M ± SD	Dieticians M ± SD	Others M ± SD	*p* ^#^
Total score	3.60 ± 0.57	3.70 ± 0.48	3.51 ± 0.48	3.44 ± 0.77	**<0.001**
Lazy/industrious	1.68 ± 0.08	3.29 ± 0.75	2.98 ± 0.67	2.96 ± 1.09	**<0.001**
No willpower/has willpower	3.16 ± 0.97	3.71 ± 0.79	3.56 ± 0.85	3.46 ± 1.09	**0.028**
Attractive/unattractive	3.68 ± 0.91	3.80 ± 0.83	3.39 ± 0.96	3.57 ± 1.14	**0.002**
Good self-control/poor self-control	3.81 ± 0.81	3.93 ± 0.72	3.80 ± 0.80	3.53 ± 1.06	**<0.001**
Fast/slow	3.67 ± 0.92	3.77 ± 0.79	3.55 ± 0.94	3.54 ± 1.14	**0.048**
Having endurance/having no endurance	3.45 ± 0.93	3.57 ± 0.78	3.43 ± 0.96	3.16 ± 1.09	**<0.001**
Active/inactive	3.57 ± 0.91	3.73 ± 0.86	3.41 ± 0.87	3.31 ± 1.13	**<0.001**
Weak/strong	3.00 ± 0.86	3.09 ± 0.74	2.86 ± 0.90	2.89 ± 1.05	**0.008**
Self-indulgent/self-sacrificing	3.41 ±0.97	3.50 ± 0.88	3.27 ± 0.97	3.32 ± 1.15	**0.032**
Dislikes food/likes food	4.13 ± 0.89	4.18 ± 0.86	4.04 ± 0.95	4.06 ± 0.89	0.164
Shapeless/shapely	3.46 ± 1.04	3.56 ± 0.94	3.10 ± 1.12	3.45 ± 1.17	**<0.001**
Undereats/overeats	4.09 ± 0.87	4.21 ± 0.81	3.95 ± 0.97	3.90 ± 0.91	**0.002**
Insecure/secure	3.76 ± 0.93	3.65 ± 0.92	3.83 ± 0.88	3.38 ± 1.05	**0.001**
Low self-esteem/high self-esteem	3.76 ± 0.93	3.79 ± 0.89	3.94 ± 0.89	3.57 ± 1.00	**0.004**

Items 3, 4, 5, 6, 7, 10, and 12 scored as follows: 1 2 3 4 5; Items 1, 2, 8, 9, 11, 13, and 14 scored as follows: 5 4 3 2 1. Fat Phobia Scale (FPS) score of the overweight vignette was from 1 = positive attributes to 5 = negative attributes. M—mean, SD—Standard deviation, # Kruskal–Wallis H Test, Significant effects (<0.05) are marked in bold.

**Table 3 nutrients-16-00999-t003:** The analysis of correlation between the FPS score and the likelihood of social interactions with individuals with obesity.

	Probability							
	Employment of a Person with Obesity		Going on a Date with a Person with Obesity		Entrusting the Care of Your Children to a Person with Obesity		Befriending a Person with Obesity	
	r	*p*	r	*p*	r	*p*	r	*p*
**FPS total score**	−0.151	**<0.001**	−0.310	**<0.001**	−0.202	**<0.001**	−0.122	**0.001**

Significant effects (<0.05) are marked in bold.

**Table 4 nutrients-16-00999-t004:** Custom questions assessing the level of stigmatization of patients with obesity.

Analyzed Variables		Studied Group N(%)/M ± SD		Healthcare Professionals		
			Physicians (%)/M ± SD	Dieticians (%)/M ± SD	Others (%)/M ± SD	*p*
Are people with obesity worse than people with normal weight?	Definitely yes	5 (0.7)	4 (1.0)	0 (0.0)	1 (0.6)	**0.011 ***
	Yes	11 (1.6)	4 (1.0)	3 (2.5)	4 (2.4)	
	Rather yes	24 (3.5)	12 (3.0)	1 (0.9)	11 (6.7)	
	Rather no	78 (11.3)	46 (11.4)	10 (8.6)	22 (13.1)	
	No	231 (33.6)	152 (37.6)	30 (25.6)	49 (29.3)	
	Definitely no	339 (49.3)	186 (46.0)	73 (62.4)	80 (47.9)	
Are people with obesity less attractive than people with normal weight?	Definitely yes	99 (14.4)	60 (14.9)	11 (9.4)	28 (16.7)	**<0.001 ***
	Yes	141 (20.5)	86 (21.3)	17 (14.5)	38 (22.8)	
	Rather yes	273 (39.7)	177 (43.8)	50 (42.7)	46 (27.5)	
	Rather no	83 (12.1)	43 (10.6)	14 (12.0)	26 (15.6)	
	No	59 (8.6)	26 (5.4)	13 (11.1)	20 (12.0)	
	Definitely no	33 (4.7)	12 (3.0)	12 (10.3)	9 (5.4)	
Would you hire a person with obesity as an employer?	Definitely yes	175 (25.5)	102 (25.3)	32 (27.4)	41 (24.6)	**<0.001 ***
	Yes	312 (45.3)	204 (50.4)	45 (38.5)	63 (37.7)	
	Rather yes	151 (21.9)	81 (20.1)	22 (18.8)	48 (28.7)	
	Rather no	38 (5.5)	12 (3.0)	12 (10.3)	14 (8.4)	
	No	7 (1.1)	2 (0.5)	4 (3.3)	1 (0.6)	
	Definitely no	5 (0.7)	3 (0.7)	2 (1.7)	0 (0.0)	
Is obesity a cause for shame?	Definitely yes	14 (2.0)	9 (2.2)	1 (0.9)	4 (2.4)	**<0.001 ***
	Yes	36 (5.2)	28 (6.9)	1 (0.9)	7 (4.2)	
	Rather yes	156 (22.7)	99 (24.6)	14 (12.0)	43 (25.8)	
	Rather no	195 (28.3)	116 (28.7)	27 (23.0)	52 (31.1)	
	No	185 (26.9)	108 (26.7)	40 (34.2)	37 (22.1)	
	Definitely no	102 (14.9)	44 (10.9)	34 (29.0)	24 (14.4)	
On a scale of 1 to 10, how likely is it in your case?						
Employment of a person with obesity		8.11 ± 2.29	8.34 ± 2.08	7.81 ± 2.46	7.79 ± 2.59	**0.039 #**
Going on a date with a person with obesity		4.47 ± 2.89	4.36 ± 2.87	4.42 ± 2.83	4.78 ± 2.98	0.315 #
Entrusting the care of your children to a person with obesity		7.88 ± 2.43	7.93 ± 2.30	7.88 ± 2.57	7.75 ± 2.67	0.847 #
Befriending a person with obesity		8.91 ± 1.94	8.94 ± 1.80	9.07 ± 1.88	8.75 ± 2.29	0.479 #
People with obesity are discriminated against in Poland		7.62 ± 2.27	7.52 ± 2.28	7.52 ± 2.26	7.94 ± 2.25	**0.038 #**
Feelings accompanying interactions with individuals living with obesity						
Mercy		59 (8.6)	34 (8.4)	9 (7.7)	16 (9.6)	0.841 *
Reluctance		60 (8.7)	43 (10.6)	3 (2.6)	14 (8.4))	**0.009 ***
Contempt		18 (2.6)	11 (2.7)	0 (0.0)	7 (4.2)	**0.022 ***
Kindness		326 (47.4)	179 (44.3)	76 (64.9)	71 (42.5)	**<0.001 ***
Discomfort in contacts		85 (12.4)	56 (13.9)	9 (7.7)	20 (12.0)	0.173 *
Liking/affection		162 (23.5)	81 (20.1)	37 (31.6)	44 (26.4)	**0.023 ***
Impatience		71 (10.3)	60 (14.9)	3 (2.6)	8 (4.8)	**<0.001 ***
Indifference		133 (19.3)	72 (17.8)	20 (17.1)	41 (24.6)	0.155 *
Compassion		260 (37.8)	170 (42.1)	41 (35.0)	49 (29.3)	**0.012 ***
Willingness to help		392 (57.0)	257 (63.6)	83 (70.9)	52 (31.1)	**<0.001 ***

M—mean, SD—Standard deviation, N—number, # Kruskal–Wallis H Test, * Chi-squared test, Significant effects (<0.05) are marked in bold.

**Table 5 nutrients-16-00999-t005:** Analysis of the differences in the FPS total score and engagement in positive social interactions by demographic and weight.

Analyzed Variables		FPS Total Score		Probability of Employment of a Person with Obesity		Probability of Going on a Date with a Person with Obesity		Probability of Entrusting the Care of Your Children to a Person with Obesity		Probability of Befriending a Person with Obesity		People with Obesity Are Discriminated in Poland	
		M ± SD	*p*	M ± SD	*p*	M ± SD	*p*	M ± SD	*p*	M ± SD	*p*	M ± SD	*p*
Sex	Female	3.59 ± 0.57	0.204 *	8.29 ± 2.20	**<0.001 ***	4.67 ± 2.88	**<0.001 ***	7.99 ± 2.39	**<0.001 ***	9.01 ± 1.88	**<0.001 ***	7.81 ± 2.11	**<0.001 ***
	Male	3.69 ± 0.58		6.81 ± 2.57		2.94 ± 2.49		7.02 ± 2.63		8.23 ± 2.30		6.23 ± 2.89	
BMI	Underweight	3.62 ± 0.66	0.674 #	7.69 ± 1.95	**0.046 #**	3.57 ± 2.25	**<0.001 #**	7.20 ± 2.29	**0.001 #**	9.11 ± 1.28	0.497 #	8.00 ± 1.98	**<0.001 #**
	Normal weight	3.61 ± 0.53		8.08 ± 2.36		3.93 ± 2.71		7.83 ± 2.49		8.85 ± 2.00		7.34 ± 2.32	
	Overweight	3.61 ± 0.68		8.12 ± 2.14		4.64 ± 2.73		7.79 ± 2.34		8.91 ± 2.00		7.73 ± 2.24	
	Obesity	3.55 ± 0.60		8.42 ± 2.30		6.69 ± 2.86		8.38 ± 2.32		9.14 ± 1.83		8.48 ± 1.99	

M—mean, SD—Standard deviation, # Kruskal–Wallis Test, * Mann–Whitney U, FPS—Fat Phobia Scale, Significant effects (<0.05) are marked in bold.

**Table 6 nutrients-16-00999-t006:** Analysis of associations between feelings accompanying interactions with individuals living with obesity and demographic and weight characteristics.

Feelings Accompanying Interactions with Individuals Living with Obesity	Sex *n* (%)			BMI *n* (%)				
	Male	Female	*p*	Underweight	Normal Weight	Overweight	Obesity	*p*
Mercy	9 (11.1)	50 (8.3)	0.657 *	4 (11.4)	38 (9.1)	10 (7.6)	7 (6.6)	0.752 *
Reluctance	12 (14.8)	48 (7.9)	0.114 *	2 (5.7)	44 (10.6)	12 (9.2)	2 (1.9)	**0.013 ***
Contempt	6 (7.4)	12 (2.0)	**0.048 ***	1 (2.9)	10 (2.4)	4 (3.1)	3 (2.8)	0.987 *
Kindness	27 (33.3)	299 (49.3)	**0.016 ***	18 (51.4)	190 (45.7)	55 (42.0)	63 (59.4)	**0.037 ***
Discomfort in contacts	13 (16.1)	72 (11.9)	0.521 *	6 (17.1)	53 (12.7)	20 (15.3)	6 (5.7)	0.101 *
Liking/affection	12 (14.8)	149 (24.6)	**0.029 ***	7 (20.0)	95 (22.8)	31 (23.7)	29 (27.4)	0.748 *
Impatience	12 (14.8)	59 (9.8)	0.367 *	2 (5.7)	47 (11.3)	16 (12.2)	6 (5.7)	0.238 *
Indifference	28 (34.6)	105 (17.3)	**<0.001 ***	5 (14.3)	85 (20.4)	28 (21.4)	15 (14.2)	0.381 *
Compassion	25 (30.9)	235 (38.8)	0.283 *	6 (17.1)	152 (36.5)	56 (42.8)	46 (43.4)	**0.017 ***
Willingness to help	42 (51.9)	349 (57.6)	0.353 *	24 (68.6)	237 (57.0)	72 (55.0)	59 (55.7)	0.529 *

BMI—body mass index; N—number; * Chi-squared test; Significant effects (<0.05) are marked in bold.

**Table 7 nutrients-16-00999-t007:** A detailed overview of the level of knowledge about obesity.

Analysed Variables		Studied Group N(%)	Healthcare Professionals			
			Physicians (%)/M ± SD	Dieticians (%)/M ± SD	Others (%)/M ± SD	*p* *
Obesity	is a chronic disease that requires treatment	678 (98.5)	400 (99.0)	114 (97.4)	164 (98.2)	0.501 *
	is a disease that does not require treatment	2 (0.3)	1 (0.3)	0 (0.0)	1 (0.6)	
	is not a chronic disease	8 (1.2)	3 (0.7)	3 (2.6)	2 (1.2)	
BMI criterion for the diagnosis of obesity	≥25	41 (6.0)	7 (1.7)	6 (5.1)	28 (16.8)	**<0.001 ***
	≥27.5	39 (5.7)	16 (4.0)	2 (1.7)	21 (12.6)	
	≥30	583 (84.7)	367 (90.8)	108 (92.3)	108 (64.6)	
	≥29	25 (3.6)	14 (3.5)	1 (0.9)	10 (6.0)	
Causes of obesity	Lack of physical activity	649 (94.3)	386 (95.5)	110 (94.0)	153 (91.6)	0.198 *
	Excessive calorie supply	680 (98.8)	404 (100.0)	116 (99.2)	160 (95.8)	**<0.001 ***
	Certain chronic diseases, e.g., diabetes	494 (71.8)	271 (67.1)	76 (65.0)	147 (88.0)	**<0.001 ***
	Hyperthyroidism	182 (26.5)	58 (14.4)	27 (23.1)	97 (58.1)	**<0.001 ***
	Effects of certain drugs, e.g., steroids and antipsychotics	626 (91.0)	375 (92.8)	96 (82.1)	155 (92.8)	**0.003 ***
Effective form of obesity treatment	Bariatric surgery	589 (85.6)	376 (93.1)	99 (75.2)	125 (74.9)	**<0.001 ***
	Pharmacological treatment	599 (87.1)	378 (93.6)	88 (75.2)	133 (79.6)	**<0.001 ***
	Regular physical activity	671 (97.5)	397 (98.3)	113 (96.6)	161 (96.4)	0.332 *
	Starvation	28 (4.1)	16 (4.0)	1 (0.9)	11 (6.6)	**0.032 ***
Obesity complications	Poorer exercise tolerance/fatigue	679 (98.7)	404 (100.0)	115 (98.3)	160 (95.8)	**<0.001 ***
	Hypertension	679 (98.7)	401 (99.3)	117 (100.0)	161 (96.4)	**0.012 ***
	Diabetes mellitus	677 (98.4)	403 (99.8)	116 (99.2)	158 (94.6)	**<0.001 ***
	Hormonal disorders	660 (95.9)	396 (98.0)	115 (98.3)	149 (89.2)	**<0.001 ***
	Female menstrual disorders	639 (92.9)	394 (97.5)	108 (92.3)	137 (82.0)	**<0.001 ***
	Decrease in libido	641 (93.2)	387 (95.8)	112 (95.7)	142 (85.0)	**<0.001 ***
	Deterioration of hair and/or nail growth	514 (74.7)	308 (76.2)	96 (82.1)	110 (65.9)	**0.005 ***
	Depression	677 (98.4)	401 (99.3)	117 (100.0)	159 (95.2)	**0.002 ***
	Future dementia	437 (63.5)	265 (65.6)	91 (77.8)	81 (48.5)	**<0.001 ***
	Gout	493 (71.7)	315 (78.0)	99 (84.6)	79 (47.3)	**<0.001 ***

Significant effects (<0.05) are marked in bold, * Chi-squared test.

## Data Availability

The database used in this study is available upon request. The data are not publicly available due to privacy restrictions.

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
