# Peer review of "Is Obesity a Cause for Shame? Weight Bias and Stigma among Physicians, Dietitians, and Other Healthcare Professionals in Poland—A Cross-Sectional Study"

_nutrients, 2024, doi:10.3390/nu16070999_

Round 1

Reviewer 1 Report

Comments and Suggestions for Authors

This is an interesting paper, the study is relevant, and the authors contribute to this field of research.

The title reflects the main purpose of the study, but I propose a small change in the title of the manuscript, to make it clearer: The whole article is very well done, congratulations to the authors

Author Response

The authors would like to thank the Reviewer for the positive feedback and a comment that helped to improve the quality of the paper. Our response is listed below.

COMMENT #1.1: This is an interesting paper, the study is relevant, and the authors contribute to this field of research. The title reflects the main purpose of the study, but I propose a small change in the title of the manuscript, to make it clearer: The whole article is very well done, congratulations to the authors.

RESPONSE #1.1: Thank you very much for your comment and positive feedback. We made an amendment in the title clarifying the type of study. The new proposed title is: “Is obesity a cause for shame? Weight bias and stigma among physicians, dietitians and other healthcare professionals in Poland – a cross-sectional study”.

Reviewer 2 Report

Comments and Suggestions for Authors

 General comments.  This paper addresses weight bias among health care providers.  While the topic is reasonably well studied, the study population is unique.  In addition, the importance of the topic and likely dynamic nature of weight bias in health care warrants continued and ongoing research in this area.

The aim of the paper "to evaluate the level of weight bias and stigma among health care professionals...in Poland," is quite broad and does not give a clear picture of the purpose or help direct the analysis.  As a result the manuscript is quite long and at times difficult to discern the points the authors are making.  

Instruments:  More detail on the data collection instruments is needed.-- Presumably the Fat Phobia scale was translated to Polish.  How and by whom was this done?  How was the translation validated?   The Fat Phobia Scale has a summary score with levels of phobia categorized and validated.  The validity of applying those same categories of phobia to each individual item on the scale has not been addressed and probably is not appropriate.  For example, I don't think the authors can say (line 167 -169) that no individual score exceeded 4.4 signaling a high level of fat phobia....without presenting the validity of using individual scores similarly as the summary score.  Excluding referencing specific categories of bias for each item score would not affect the manuscript quality though.

The third and fourth sections contain custom questions regarding social interactions and knowledge of obesity causes that may be indicative of weight stigma. This is a big topic with many options on questions, therefore, it is important to know how and why the questions selected.  

Statistical analysis.  The statical approaches are appropriate for a descriptive study. However, there are too many 2x 3, x4, comparisons. In some cases, multivariable analyses would be more informative.

Results

There is a lot of data which could be summarized in a more succinct and informative way. 

Table 2:  The purpose of table 2 is to compare  health care professionals on the composite score and each scale item.   I think "Final Score" would be better labeled as total score or composite score and listed first on the table---that is the way it is presented in the text and, to me,  is the most important result.  Physicians scored higher on the composite Fat Phobia score as well as all the individual items- A key  point that was not presented in the text was that physicians were consistently higher in all individual items.  Since comparing weight bias across health care professional groups seems to be one purpose of the study, it is important to emphasize this.

I don't know who is being referred to by "both consecutive groups" line 176 

Table 3 .  The important point here is that there was a significant, but weak, inverse correlation in all but 12 of the 64 correlations.  Presenting the strength of the association helps in the interpretation---The strongest association was observed between the composite score and dating.  I am actually not sure how presenting all the individual correlations advances knowledge or science. 

Table 4.  It would be helpful to collapse the categories.  Five categories are too difficult to interpret in any meaningful way.  For example what is the difference or significance  between Definitely yes and yes?

The inclusion of the "feeling" variables needs more justification or explanation in the methods section.

Table 5.   Table 5 is mislabeled.  It is not examining the impact of demographic variables on the composite Fat Phobia Scale, rather it is examining the differences in fat phobia scores and social variables by demographic and weight characteristics.  It is not clear what point the authors are trying to make with this table.  A multivariable analyses could be helpful here.

Table 6.   This table is mislabeled. First the FBS score is not anywhere on the table and as with table 5, this table presents differences in "custom questions" by sex and BMI. Other than sex, there are no demographic variables included.  The "custom questions" are different for both analyses (table 5 and 6) and is therefore confusing.  Secondly, the importance of the table is not clear since there is no explanation or justification for the inclusion of these custom variables in either the introduction or methods.

The first 1 1/2 paragraphs of the discussion repeats the introduction or reports on the authors previous studies and can be eliminated.  This would help get to the important points and shorten a long discussion.

The conclusions overstate the findings with respect to informing interventions.  

Author Response

The authors would like to thank the Reviewer for the comments that helped to improve the quality of the paper. We have carefully analysed all of them and modified the manuscript. Detailed responses to individual comments are listed below.

COMMENT #2.1:  General comments.  This paper addresses weight bias among health care providers.  While the topic is reasonably well studied, the study population is unique.  In addition, the importance of the topic and likely dynamic nature of weight bias in health care warrants continued and ongoing research in this area. The aim of the paper "to evaluate the level of weight bias and stigma among health care professionals...in Poland," is quite broad and does not give a clear picture of the purpose or help direct the analysis.  As a result the manuscript is quite long and at times difficult to discern the points the authors are making.  

RESPONSE #2.1: Thank you for your thoughtful comments. We appreciate your recognition of the unique study population and the importance of addressing weight bias among healthcare providers. We acknowledge the need for continued research in this area given its dynamic nature. The aim of the paper is indeed broad, reflecting its pioneering nature as the first of its kind in Poland. We see it is as a foundational introduction to future, more detailed research in this field. We hope that the amendments introduced, thanks to your insightful comments, have enhanced the clarity and consistency of the manuscript.

COMMENT #2.2:  Instruments:  More detail on the data collection instruments is needed.-- Presumably the Fat Phobia scale was translated to Polish.  How and by whom was this done?  How was the translation validated?  

RESPONSE #2.2: The FPS was translated into Polish by a team member certified as bilingual. Subsequently, the Polish translation was independently back-translated into English by two experts, medical professionals proficient in English, from the Centre of Postgraduate Medical Education. Our research team then verified the conformity of the original FPS scores with their back translations and no significant differences were reported.

We added a following sentence to the “Material and methods” section: “The FPS was translated into Polish by a certified bilingual team member. It was then independently back-translated into English by two medical professionals proficient in English. The original FPS was verified against the back translations, finding no significant differences.”

COMMENT #2.3:  The Fat Phobia Scale has a summary score with levels of phobia categorized and validated.  The validity of applying those same categories of phobia to each individual item on the scale has not been addressed and probably is not appropriate.  For example, I don't think the authors can say (line 167 -169) that no individual score exceeded 4.4 signaling a high level of fat phobia....without presenting the validity of using individual scores similarly as the summary score.  Excluding referencing specific categories of bias for each item score would not affect the manuscript quality though.

RESPONSE #2.3: Thank you very much for drawing our attention to this important issue. We have removed this sentence from the manuscript.

COMMENT #2.4:  The third and fourth sections contain custom questions regarding social interactions and knowledge of obesity causes that may be indicative of weight stigma. This is a big topic with many options on questions, therefore, it is important to know how and why the questions selected.  

RESPONSE #2.4: The custom questions in both sections were developed by our multidisciplinary team of authors, drawing on their many years of expertise in obesity treatment, medical education, public health, medical communications, patients advocacy and extensive literature review.

A following sentence was added to the “Materials and methods” section: “The custom questions in both sections were developed by our multidisciplinary team of authors, drawing on their many years of expertise in obesity treatment, public health, medical education, medical communications, patients advocacy and extensive literature review.”

COMMENT #2.5:  Statistical analysis.  The statical approaches are appropriate for a descriptive study. However, there are too many 2x 3, x4, comparisons. In some cases, multivariable analyses would be more informative.

RESPONSE #2.5: Thank you for this comment. We have removed non-statistically significant data from Table 5 (place of residence variable) to improve the clarity of the presentation. While we considered conducting multivariable analyses following your suggestion, it would require new calculations. After consulting a statistician, we decided to retain the analysis  in its initial form.

COMMENT #2.6:  Results: There is a lot of data which could be summarized in a more succinct and informative way. 

RESPONSE #2.6: Thank you for this suggestion. We have implemented several changes in the “Results” section, including shortening the text and tables. These modifications are detailed in the responses to the comments below and are clearly marked in the manuscript. We believe these changes have improved the overall clarity of the manuscript.

COMMENT #2.7:  Table 2:  The purpose of table 2 is to compare  health care professionals on the composite score and each scale item.   I think "Final Score" would be better labeled as total score or composite score and listed first on the table---that is the way it is presented in the text and, to me,  is the most important result.  Physicians scored higher on the composite Fat Phobia score as well as all the individual items- A key  point that was not presented in the text was that physicians were consistently higher in all individual items.  Since comparing weight bias across health care professional groups seems to be one purpose of the study, it is important to emphasize this.

RESPONSE #2.7: Thank you for this comment. We have replaced the expression “final score” with “total score” in Table 2, as well as in other tables and throughout the text. Additionally, we have moved it to the top of the table as suggested.  In the description, we have also included information about physicians scoring higher on all individuals scale items: “When comparing the level of fat phobia among various groups of healthcare professionals, physicians scored higher on all the individual items and exhibited the highest level of explicit weight bias (…)”.

COMMENT #2.8:  I don't know who is being referred to by "both consecutive groups" line 176 

RESPONSE #2.8: To enhance clarity “both consecutive groups” were replaced by “both the group of dietitians and the group of other healthcare professionals”.

COMMENT #2.9:  Table 3 .  The important point here is that there was a significant, but weak, inverse correlation in all but 12 of the 64 correlations.  Presenting the strength of the association helps in the interpretation---The strongest association was observed between the composite score and dating.  I am actually not sure how presenting all the individual correlations advances knowledge or science. 

RESPONSE #2.9: Thank you for this comment. We concur that presenting individual correlations does not significantly contribute to knowledge beyond the FPS total score alone. As such, we have shortened the table and removed the adjective pairs, retaining only the FPS total score. Additionally, we have added the following statement to the results description: The strongest association was observed for probability of going on a date with an individual living with obesity.

COMMENT #2.10:  Table 4.  It would be helpful to collapse the categories.  Five categories are too difficult to interpret in any meaningful way.  For example what is the difference or significance  between Definitely yes and yes?

RESPONSE #2.10: The authors discussed the data aggregation methods within their group and opted to maintain the categories consistent with how distractors were presented to respondents, utilizing a classic 5-point Likert scale. Our decision to maintain the five categories aligns with standard research practices that we have followed in our previous works, ensuring consistency with how the data was collected as well as providing a more nuanced understanding of respondent perceptions. However, we acknowledge the suggestion and will consider alternative methods for future studies to enhance interpretability without compromising data integrity.

COMMENT #2.11:  The inclusion of the "feeling" variables needs more justification or explanation in the methods section.

RESPONSE #2.11: The custom questions in both sections were developed by our multidisciplinary team of authors, drawing on their many years of expertise in obesity treatment, public health, medical communications, patients advocacy and extensive literature review. The particular part related to the feelings was partially based on a recent patients’ initiative – a report focused on the experiences of individuals living with obesity with healthcare system in Poland.

A following sentence was added to the “Materials and methods” section: “The custom questions in both sections were developed by our multidisciplinary team of authors, drawing on their many years of expertise in obesity treatment, public health, medical communications, patients advocacy and extensive literature review.”

COMMENT #2.12:  Table 5.   Table 5 is mislabeled.  It is not examining the impact of demographic variables on the composite Fat Phobia Scale, rather it is examining the differences in fat phobia scores and social variables by demographic and weight characteristics.  It is not clear what point the authors are trying to make with this table.  A multivariable analyses could be helpful here.

RESPONSE #2.12: The description of the Table 5 was revised to:  “Analysis of the differences in the FPS total score and engagement in positive social interactions by demographic and weight characteristics”. The purpose of this table is to present associations between respondents’ sex/BMI and FPS score/willingness to engage in positive social interactions with individuals living with obesity. A literature review has indicated correlations between demographic and weight characteristics and the level of fat phobia or discriminatory behaviours, prompting our exploration of these associations among our respondents.

COMMENT #2.13:  Table 6.   This table is mislabeled. First the FBS score is not anywhere on the table and as with table 5, this table presents differences in "custom questions" by sex and BMI. Other than sex, there are no demographic variables included.  The "custom questions" are different for both analyses (table 5 and 6) and is therefore confusing.  Secondly, the importance of the table is not clear since there is no explanation or justification for the inclusion of these custom variables in either the introduction or methods.

RESPONSE #2.13: Thank you for this comment. The description of the Table 6 was revised to: “Analysis of the associations between feelings accompanying interactions with individuals living with obesity and demographic and weight characteristics”.

Both tables (Table 5 and 6) present associations between respondents’ sex/BMI and our measures of weight stigma (FPS and custom questions). To enhance clarity, the data was divided into two tables.

COMMENT #2.14:  The first 1 1/2 paragraphs of the discussion repeats the introduction or reports on the authors previous studies and can be eliminated.  This would help get to the important points and shorten a long discussion.

RESPONSE #2.14: Thank you for this valuable comment. We have carefully revised the initial  paragraphs of the introduction, removing redundant information and description of our previous study. This necessitated a reorganization of subsequent paragraphs. Overall, these revisions have resulted in a reduction of 494 words in the “Results” section.

COMMENT #2.15:  The conclusions overstate the findings with respect to informing interventions.  

RESPONSE #2.15: Thank you for this comment. We have revised the “Conclusions” section and incorporated “Recommendations”, where we advocate for utilizing our findings while encouraging further research in this field. While our study is the first of its kind among Polish healthcare professionals and significantly contributes to knowledge in this area, we acknowledge the study’s limitations and the necessity for additional research.

Reviewer 3 Report

Comments and Suggestions for Authors

Thank you for the opportunity to review such an interesting and impactful manuscript. The manuscript is very well-written and the inclusion of dietitians and other healthcare providers in addition to physicians is a major strength. I have outlined a few suggestions to improve the manuscript below. Great work! 

Introduction

Lines 41 – 44

In order to support the statement that there is “a mounting body of evidence,” it would be helpful to include several references to support the following statement:

“There is a mounting body of evidence that demonstrates how weight bias and stigma pose a significant challenge to the effective treatment and prevention of obesity. This challenge stems from their detrimental impact on health and their widespread prevalence, including the healthcare settings [4].”

Lines 64 – 66

It would be helpful to clarify what the subject is (what is meant by “It”), by rephrasing the following statement:

It is visible for example in public campaigns and policies that tend to overlook environmental and societal factors that play a major role in the epidemic of obesity, which in turn can further perpetuate obesity stigma among individuals [10].”

For example, the authors could replace “It” with “stigmatizing language around obesity.”

Discussion

441 – 449

While the limitations description is appropriate and there may have been improved candidness with the anonymous survey, the self-reported measures of fat phobia/obesity bias in this study (as opposed to implicit bias) are still subject to social desirability bias (e.g., responses may not align with behavior / implicit attitudes): 

"The authors acknowledge the inherent limitations associated with the methodology employed in this study, particularly regarding the chosen data collection methods. However, leveraging online platforms and social media substantially aids in reaching a diverse audience across Poland."

Comments on the Quality of English Language

Very well-written! There are a few edits that could improve the readability, but they are very minor and infrequent. 

Author Response

The authors would like to thank the Reviewer for the comments that helped to improve the quality of the paper. We have carefully analysed all of them and modified the manuscript. Detailed responses to individual comments are listed below.

COMMENT #3.1:  Thank you for the opportunity to review such an interesting and impactful manuscript. The manuscript is very well-written and the inclusion of dietitians and other healthcare providers in addition to physicians is a major strength. I have outlined a few suggestions to improve the manuscript below. Great work! 

RESPONSE #3.1: Thank you for such a positive feedback and valuable suggestions on improving the manuscript; each point raised was addressed – as explained below.

COMMENT #3.2:  Introduction; Lines 41 – 44: In order to support the statement that there is “a mounting body of evidence,” it would be helpful to include several references to support the following statement: “There is a mounting body of evidence that demonstrates how weight bias and stigma pose a significant challenge to the effective treatment and prevention of obesity. This challenge stems from their detrimental impact on health and their widespread prevalence, including the healthcare settings [4].”

RESPONSE #3.2: Thank you for this suggestion. We have expanded the references supporting the statement ([4-10] instead of [4]) and each reference from [4-10] is individually cited in the subsequent paragraph.

It’s noteworthy that several of these references are systematic reviews, which substantiate our use of the term “mounting body of evidence”. To enhance clarity regarding the connection between the paragraph at lines 41-43 and the references cited, we’ve opted to merge it with the following paragraph (lines 44-66).

COMMENT #3.3:  Lines 64 – 66: It would be helpful to clarify what the subject is (what is meant by “It”), by rephrasing the following statement: It is visible for example in public campaigns and policies that tend to overlook environmental and societal factors that play a major role in the epidemic of obesity, which in turn can further perpetuate obesity stigma among individuals [10].” For example, the authors could replace “It” with “stigmatizing language around obesity.”

RESPONSE #3.3: Thank you for drawing attention to this sentence. Indeed, the subject was not clear. We replaced “it” with “The impact of this assumption”.

COMMENT #3.4:  Discussion - 441 – 449: While the limitations description is appropriate and there may have been improved candidness with the anonymous survey, the self-reported measures of fat phobia/obesity bias in this study (as opposed to implicit bias) are still subject to social desirability bias (e.g., responses may not align with behavior / implicit attitudes): 

"The authors acknowledge the inherent limitations associated with the methodology employed in this study, particularly regarding the chosen data collection methods. However, leveraging online platforms and social media substantially aids in reaching a diverse audience across Poland."

RESPONSE #3.4: Thank you for this insightful comment. We added a remark regarding social desirability bias in the sentence as follows: “The utilization of an anonymous internet survey format might encourage more candid responses, especially given the sensitivity of the subject matter, although it does not exclude the social desirability bias.”

Round 2

Reviewer 2 Report

Comments and Suggestions for Authors

The critical concerns have been addressed

Reviewer 3 Report

Comments and Suggestions for Authors

The authors have fully addressed all of my previous comments, and I must express my appreciation for the professional and polite tone of their response. I have no further suggestions for the authors and wish them the best of luck moving forward.